# The Role of *Nosema ceranae* (Microsporidia: Nosematidae) in Honey Bee Colony Losses and Current Insights on Treatment

**DOI:** 10.3390/vetsci9030130

**Published:** 2022-03-11

**Authors:** Pablo Jesús Marín-García, Yoorana Peyre, Ana Elena Ahuir-Baraja, María Magdalena Garijo, Lola Llobat

**Affiliations:** 1Department of Animal Production and Health, Veterinary Public Health and Food Science and Technology (PASAPTA), Facultad de Veterinaria, Universidad Cardenal Herrera-CEU, CEU Universities, 46115 Valencia, Spain; pablo.maringarcia@uchceu.es (P.J.M.-G.); ana.ahuir@uchceu.es (A.E.A.-B.); 2Facultad de Veterinaria, Universidad Cardenal Herrera-CEU, CEU Universities, 46115 Valencia, Spain; peyre.yoorana@gmail.com

**Keywords:** *Apis mellifera*, *Nosema ceranae*, nosemosis, colony losses

## Abstract

Honeybee populations have locally and temporally declined in the last few years because of both biotic and abiotic factors. Among the latter, one of the most important reasons is infection by the microsporidia *Nosema ceranae*, which is the etiological agent of type C nosemosis. This species was first described in Asian honeybees (*Apis cerana*). Nowadays, domestic honeybees (*Apis mellifera*) worldwide are also becoming infected due to globalization. Type C nosemosis can be asymptomatic or can cause important damage to bees, such as changes in temporal polyethism, energy and oxidative stress, immunity loss, and decreased average life expectancy. It causes drastic reductions in workers, numbers of broods, and honey production, finally leading to colony loss. Common treatment is based on fumagillin, an antibiotic with side effects and relatively poor efficiency, which is banned in the European Union. Natural products, probiotics, food supplements, nutraceuticals, and other veterinary drugs are currently under study and might represent alternative treatments. Prophylaxis and management of affected colonies are essential to control the disease. While *N. ceranae* is one potential cause of bee losses in a colony, other factors must also be considered, especially synergies between microsporidia and the use of insecticides.

## 1. Introduction

Global agricultural production requires entomophile pollination. In 2005, pollination by bees represented 9.5% of the world agricultural production destined for human consumption. The value fluctuated among different countries, between 1.8% (Turkey) and 53% (Ireland) [1,2]. In fact, a recent review by Wagner indicates that declines of insect populations, including honeybees (*Apis mellifera*), have occurred mainly in Europe due to biotic and abiotic factors, such as agricultural intensification, climate change, and invasive species [3]. These local and temporal declines are product of socioeconomic transformations [4]. The COLOSS Honeybee Research Association collects data from more than 30 countries, most of them in the Northern Hemisphere, to monitor the loss of bees around the world on an annual basis [5]. The last results obtained by COLOSS indicated an overall loss rate of 20.9% in the winter of 2016/2017, including data of 425,762 colonies in 30 countries around the world [6] with standard protocols [7]. Possible causes for the colony losses include abiotic and biotic factors [8]. Abiotic factors, such as climate change, forage shortages, or uncontrolled use of chemicals, insecticides, and herbicides have been shown to have a strong influence [9,10,11,12,13,14]. The biotic factors include different species of acarids (*Varroa destructor* and *Acarapis woodi*), insects (*Aethina tumida*), viruses of the family Dicistroviridae, and bacteria (*Paenibacillus larvae* and *Melissococcus plutonius*) [15,16,17,18]. Combined effects derived from abiotic and biotic factors may increase bee population losses [19,20,21,22]. 

The second most prevalent biological agents related to the decrease in worker bees are the parasitic microsporidia *Nosema* spp., mainly *Nosema apis* and *N. ceranae*, both responsible for nosemosis in *Apis* spp. [21]. *N. ceranae* was first described in the Asian honeybee (*Apis cerana*), and it replaced *N. apis*, although not generally, in European honeybees (*A. mellifera*) around 3 decades ago [23,24,25,26,27,28,29,30]. According to a study by Gisder et al. (2017), *N. apis* and *N. ceranae* show different multiplication rates in cell culture, but this is possibly not relevant in vivo [24]. Even though *N. apis* and *N. ceranae* show similar virulence, multiplication, and mortality rates [31], *N. ceranae* is more prevalent and suppresses the bee immune response, which has been related to colony collapse [32,33,34]. *N. ceranae* infections seem to show seasonality, with higher infection levels in spring, and depend on geography and other factors, such as the presence of viruses [31,35,36,37]. The main aim of this review is to deepen our understanding of the role of *N. ceranae* in the decrease of bee populations and to present new detection and diagnosis methods and current nonchemical treatments against infection. 

## 2. Etiology of *Nosema* spp. Infection

*Nosema* spp. is an intestinal microsporidian fungus spore-former (in its infecting form) and an obligate intracellular parasite of eukaryotes [38]. Microsporidia have a simplified mitochondrial form, a mitosome, which does not allow the production of ATP, implying a strong energy dependence on the host, the key to its pathogenicity [39]. Nosemosis is a parasitic disease caused by two species of *Nosema*. Type A nosemosis is caused by *N. apis* and type C is mainly caused by *N. ceranae* [12,40], causing dysentery [41]. *N. apis* and *N. ceranae* present a similar internal structure, but differ in the size of the spores, with the spores of *N. apis* being bigger than those of *N. ceranae* (6 × 3 µm vs. 4 × 2.2 µm) [33]. Traditionally, differentiation of *N. ceranae* and *N. apis* has been carried out by transmission electron microscopy (TEM) observation as the polar tube coils can be counted by using TEM [42], since the spores of *N. ceranae* have 20 to 23 spirals of polar filaments, less than *N. apis* which usually contains 30 to 44 spirals of polar filaments [27,43]. Differentiation and etiologic diagnosis under TEM require qualified staff and is labor intensive; therefore, molecular detection by PCR is needed [35]. Mixed infections can occur in East Asia and America, although they have also been reported in other geographic areas [44,45]. Type A nosemosis is opportunistic and affects already-weakened colonies. This disease is favored by prolonged inclement weather, hibernation, and certain commercial practices, which lead to confinement of the hive [46].

Type C nosemosis can be asymptomatic or can cause important damage to bees [41,47,48], showing a seasonal pattern directly related to increased temperature [24,35,49]. *Nosema ceranae* infects mainly worker bees [47,48], inducing early maturation of nurses, which causes an imbalance in the hive [48,49]. Higes et al. (2008) found a correlation between this species and honeybee colony collapse. According to these authors, hives infected by *N. ceranae* undergo what is known as an incubation phase, when the queen can produce enough offspring to compensate the loss of workers. During this long phase, clinical signs are not present, but when more than 80% of the bees are infected with more than 10 million spores, collapse occurs. At this moment, the queen cannot produce enough eggs and the number of nurses and forager bees is reduced. However, some studies have not reported evidence of clinical signs usually observed in infections of *N. apis*, such as dysentery, crawling bees, or supersedure of the queen [47,50]. 

*Nosema ceranae* was first discovered by Fries et al. (1996) in Asian honeybees (*A. cerana*), but today this species is present worldwide [27]. Since restrictions on the importation of bees and bee products in Western Australia kept the region free of nosemosis, a relationship between the parasite and commercial exchanges may be suggested [28,46]. *N. apis* has been replaced by *N. ceranae* in tropical areas, but the coexistence of both species has also been reported in cold areas, even though the viability of spores decreases by around 30% at low temperatures [31,51,52,53,54]. Besides *A. cerana*, *N. ceranae* can infect other species, such as *A. mellifera*, *A. dorsata,* and *A. florea* and bumblebee *Bombus bellicosus* [55,56,57]. These three bee species can become parasitized, and horizontal fecal–oral transmission is common [30,50,58]. Although much less important to disease spread, vertical transmission is also possible, since spores have been found in the ovaries of queens [35,59]. Young worker bees are usually infected by *N. ceranae* with fecal spots containing infective spores, and trophallaxis and food exchange between bees also cause transmission [60]. Moreover, parasites can be spread among different colonies. Substances such as honey, wax, royal jelly, and pollen have been found to act as fomites [42,51]. 

Type C nosemosis has a complex pathogenesis. This microsporidium alters the physiology and behavior of individual bees, as well as the whole hive. Depending on their age, bees develop different functions, e.g., young bees clean, build, and nurse inside the hive, whereas exterior tasks are reserved for older bees. Hormones (vitellogenin (Vg) and juvenile hormone III (JH)) are physiological regulators underlying behavioral development in bees. *N. ceranae* infection provokes a hormone imbalance, accelerating this development [60,61]. Infected queens show increased Vg titers [58], workers show higher concentrations of JH [62], and high Vg levels in younger bees are associated with *N. ceranae* infection, which can delay polyethism and disrupt colony balance [61]. Pheromones are also relevant in the stage transitions, including brood pheromone (BP), ethyl oleate (EO), (E)-β-ocimene (EBO), and queen mandibular pheromone (QMP). BP and EO are produced by young bees, whereas EBO is produced by older bees, and QMP by queen bees [63,64,65]. In infected young bees, EO levels are similar to those of noninfected older bees. This means early maturation and alteration of QMP levels, which changes the worker bees’ behavior toward the queen [58,66]. 

About effects on the metabolism, *N. ceranae* has been found to reduce food sharing between bees, suggesting an increased hunger level, which may influence the infection transmission rates [67]. Mayack and Naug (2009) confirmed that infection induces energetic stress, manifested by increased appetite [68]. In addition, alterations in amino acids, lipids, and carbohydrate reserves have been shown. This impairs the ability to fly [39,69] due to energetic stress or disorientation [70]. Furthermore, infected bees present atrophy of hypopharyngeal glands, which secrete major royal jelly proteins and glucosidase III [71]. Nosemosis also affects life expectancy, which decreases in infected bees [60]. *N. ceranae* can also affect hive behavior, with lower honey and offspring production and lower numbers of worker bees in infected hives [49,52,63]. 

The use of insecticides is a very important anthropic factor causing synergies with *N. ceranae* infection. The first evidence of potential synergy between insecticides and *N. ceranae* infection was verified by Alaux et al. (2010), who showed a higher mortality rate of infected bees exposed to imidacloprid [10]. A year later, similar results were found with other insecticides, such as fipronil and thiacloprid, at sublethal doses [72]. Other studies indicated that this phenomenon was independent of the intake and number of *N. ceranae* spores, and it occurs in natural environments [73]. The synergistic effect of nosemosis with fipronil can increase mortality from 23–39% with *N. ceranae* and fipronil, separately, to 84% with the two combined [74]. Under laboratory conditions, the presence of xenobiotics alters gene expression related to immunity and decreases the survival of bees infected by *N. ceranae* [67]. Moreover, synergy between infection and insecticide exposure affects the bees’ microbiota, which harms their general health [75]. Nowadays, it is possible to evaluate the effects of sublethal exposure to insecticides in *N. ceranae* infection around the world by simulation [76]. Besides insecticides, consumption of pollen with higher fungicide loads has been demonstrated to increase bees’ susceptibility to *Nosema* infection [77]. However, synergies are evident not only with exposure to chemical stressors. Recently, Arismendi et al. (2020) studied infections of *Lotmaria passim*, a predominant trypanosomiasis in honeybees, mixed with *N. ceranae*. Their results showed lower survival rates as a consequence of this coinfection, due to a decrease in immune-related gene expression [78].

## 3. Host Resistance of *N. ceranae*

To protect themselves against pathogens, insects form a chitinous exoskeleton and a cuticle in the intestinal tract. This cuticle is not present in the middle intestine, in which only a semipermeable peritrophic membrane is present that allows the passage of pathogens [79,80]. In addition, bees produce reactive oxygen species (ROS) with antimicrobial properties [81]. This defense mechanism is inefficient against *N. ceranae* infection, as this microsporidium provokes an overexpression of genes related to oxidation, which causes oxidative stress and damage in intestinal cells by the production of enzymes such as catalase, peroxidase glutathione, and S-transferase glutathione [66]. A low level of Vg related to infection by the pathogen improves oxidative stress, but *N. ceranae* can inhibit it [81]. Some enzymes, such as prophenol-phenoloxidase (PO), dehydrogenase glucose (DG), and lysozyme (LYS), are related to cellular immunity in bees [77,82]. Meanwhile, humoral immunity involves the production of antimicrobial peptides (AMPs) such as apidaecin, abaecin, hymenoptaecin, and defensin, which act in the membrane of the pathogen [32]. Some studies have shown that *N. ceranae* infection causes underexpression of genes related to the synthesis of AMPs and the enzymes cited, inducing immunosuppression of the host, whereas the melanization process does not seem to be modified [32,69,83,84]. 

Besides individual immunity, bees present a social defense mechanism, called social immunity [85], that is also related to *Nosema* infection. This phenomenon consists of different events, such as corpse transportation, altruistic self-removal of sick individuals, grooming or behavioral fever (increasing the body temperature around a pathogen), propolis production, and secretion of antimicrobial molecules, such as glucose oxidase, in food by nurses for young bees [86]. McDonnell et al. (2013) confirmed that noninfected bees do not show aggressive behavior toward sick individuals [87]. Moreover, allogrooming has been described as an important defense mechanism against *Varroa* or *Acarapis* mites. Since grooming involves licking and chewing, it has been suggested that this is a viral strategy to increase transmission [84] and may also favor the spread of *N. ceranae*. On the other hand, bees infected with *Nosema* are known to forage precociously and often die before returning to the nest, which may be an adaptation to lower the rate of disease transmission. The ability to resist infection also depends on exogenous factors, such as the main resources available and the amount of nutrients, which are essential to compensate the energetic stress due to the disease and to boost the immune system [68].

## 4. Detection and Outcome of *N. ceranae* Infection

Detection of type C nosemosis can be carried out both in individuals and in the hive. In the hive, older bees show higher mortality rates, and problems to go back in the nest (orientation problems). This causes younger bees to reach maturity, which decreases the general life expectancy and population [51,87]. The condition of young bees is altered, with a lower number of hemolymph cells. In addition, the storage of resources is reduced (mainly for honey production) and the early replacement of the queen can be observed [88]. However, these signs are not specific to nosemosis type C; thus, a differential diagnosis becomes necessary. Furthermore, it is important to consider the absence of clinical signs at a low level of infection. Differentiation between infection by *N. apis* or *N. ceranae* can be achieved by microscopic identification of spores, although molecular identification is the method of choice [89]. In fact, the World Organization for Animal Health (OIE) recommends the use of multiplex PCR for microsporidia identification [31,48,90]. In the last years, several research groups have been working on improving different molecular techniques. Lannutti et al. (2020) developed loop-mediated isothermal amplification (LAMP) for detection of *N. ceranae* [91]. The same year, Ribani et al. confirmed that environmental DNA analysis in honey could be a useful tool for detecting *N. ceranae* [91]. Alternatively, different genetic variants of *N. ceranae* trigger different immune responses in the host [92], making diagnosis even more difficult. These detection-related handicaps could be overcome using molecular methods [93,94].

The outcome depends on the moment of detection and the infection level of each hive, with forager bees being the most reliable samples for *N. ceranae* detection, and they should be collected at the hive entrance [95]. On the other hand, an increased number of bees for analysis is important to collect the highest possible number of forager bees [96,97]. Late detection and/or high infection levels can lead to severe mortality rates and worse prognosis [88]. In its acute form, the disease provokes the trembling of honeybee workers and dead bees around the hive. The bees show a dilated abdomen and brown fecal marks on the comb and the front of the hive. Infected colonies have decreased brood production and slowed colony growth [98,99]. At the individual level, the hypopharyngeal glands of infected nurse bees lose the ability to produce royal jelly, the production of mature larvae decreases, young infected nurse bees cease brood rearing and turn to guarding and foraging duties, infected queens cease egg-laying, life expectancy is reduced, and the disease contributes to increased dysentery [100]. Longevity data are sparse, but some investigators reported 100% mortality between 10 and 14 days after spore exposure [101,102], while others reported lower mortality rates [31,64,103,104]. For these reasons, it is necessary to increase the data on mortality rates in *Nosema* infections to mitigate the impact of infection on colony viability. For this, experiments could use marked bees with and without infection and introduce them to colonies, and then analyze the rate of disappearance of infected honeybees for a better approximation [105].

## 5. Current Insights on Treatment

Fumagillin is one of the most common treatments, which is administered as a prophylactic or control treatment [106]. Both *N. apis* and *N. ceranae* are sensitive to fumagillin, an antimicrobial substance produced by *Aspergillus fumigatus*, which temporarily reduces the parasitic burden and the risk of collapse [47,104]. However, the size of treated and nontreated colonies was shown to be similar 2 months after treatment, so the probability of surviving the winter does not differ between them [107,108]. On the other hand, fumagillin seems to alter structural and metabolic proteins in honeybees that are necessary for normal cell function. *Nosema ceranae* are apparently released from the suppressive effects of fumagillin at concentrations that impact honeybee physiology [103]. Thus, sanitation of the hive is not possible if spores remain in the honey, wax, or pollen [88,104]. In addition, after more than 50 years of commercial use, residues can be detected in hive products [109], and there are concerns that *Nosema* spp. are becoming resistant to it [106]. Finally, whereas in the USA, fumagillin is the only antibiotic approved for the control of nosemosis in honeybees, its use is banned in Europe due to the presence of its residues in honey [48,110,111]. 

All the reasons have promoted the search for alternative treatments against nosemosis, and different substances have been proved to diminish or eradicate *N. ceranae* infection (Table 1). Recently, Borges et al. (2020) analyzed 10 nutraceuticals (plant extracts and metabolites obtained from plants and spices, such as oregano oil, thymol, carvacrol, naringenin, *trans*-cinnamaldehyde, tetrahydrocurcumin, sulforaphane, embelin, allyl sulfide, and hydroxytyrosol), and concluded that high concentrations of sulforaphane reduced the number of *N. ceranae* spores in 100% of the bees, but also killed all of them, making it a poor option as an alternative treatment due to its side effects [112]. 

Baffoni et al. (2016) showed that *Bifidobacterium* and *Lactobacillus* (bacteria) supplementation reduced the number of microsporidia in *A. mellifera* [113]. Similar results were found with supplementation with *Parasaccharibacter apium*, a bacterium present in the food stores and hypopharyngeal glands of worker bees and queens, which improved resistance to *Nosema* [114]. Commercial probiotics seem to have a positive effect on reducing the number of spores in colonies and producing positive physiological changes in individual bees [110]. In fact, an increase in gut microbes in bees inhibits *N. ceranae* proliferation and improves the immune response in *A. cerana* [115]. On the contrary, a poor-quality diet favors the multiplication of *N. ceranae*, as a consequence of an altered gut microbiota and immunity in bees [116]. 

Supplementation with proteins such as propolis and pollen was shown to prevent clinical signs of infection. Pollen supplementation increases the transcription of genes related to the expression of Vg and other important bioprocesses in infected bees [116,117], and administration of propolis extract decreases *N. ceranae* spore levels in honeybees [118,119]. Suwannapong et al. (2018) observed a reduction in both mortality and infection rates after oral administration of propolis in *Apis florea* [120]. They also detected elevated levels of trehalose in hypopharyngeal glands of infected bees. In another study, propolis with methanolic plant extracts was found to increase the survival rate and significantly decrease the parasitic burden [121]. Other natural compounds such as chitosan and peptidoglycan reduce *N. ceranae* infection and improve aspects of foraging behavior [122]. Other nonchemical treatments, such as formic acid (Nosestat^®^), a natural extract based on beet extract and molasses (Vitafeed Gold^®^), and phenyl salicylate, did not show a positive effect on control of *N. ceranae* [123]. A commercial dietary supplement for veterinary use based on B group vitamins (ApiHerb^®^) and a commercial drug based on oxalic acid dihydrates (Api-Bioxal^®^) have been tested with positive results [124]. Treatment with oxalic acid decreased the number of spores in an 8-day laboratory experiment, and the prevalence of infection was reduced when it was administered to free-flying colonies twice in autumn [125]. However, sperm parameters such as the count the motility, the acrosome integrity, the membrane function of sperm, and the histomorphology of seminal vesicles are affected when drones are exposed to this product [126].

Although some of these studies may look promising, more research is needed to determine the safe use of alternatives to fumagillin treatment against nosemosis. Other chemical compounds are also being studied; among them, protoporphyrin lysine has been shown to prevent the development of *Nosema* spp. spores. In addition, this chemical reduces disorders in the absorption of nutrients in infected bees and decreases the number of spores and their viability by inactivating exospores [127,128]. Finally, plant extracts have also demonstrated the ability to inhibit *Nosema* spp. development (Table 1) [127,128].

Since honey can act as a reservoir for *N. ceranae* infective spores, even at cold temperatures when stored in the colony, management by beekeepers remains one of the main factors in controlling when infection occurs. In this sense, the spores become noninfectious when honeycombs are maintained at −12 °C (or lower) for 7 days or at 33 °C for 50 days [131]. Hives with a higher parasitic load should be removed, while the shook swarm method in hives capable of surviving the infection can delay the life cycle of the parasite [88]. Prevention is also vital. In this sense, all measures aimed at reducing between-colony transmission by beekeepers spreading spores could be important to control the disease [87], and a queen replacement every 2 years is standard practice [132]. To summarize, it is necessary to integrate hive-specific measures, evaluate engagement with stakeholders linked to bee health, and recontextualize both within landscape-scale efforts, that is, to use the “one-health” approach in order to reverse the decrease of bee numbers [133]. 

## 6. Conclusions

The role of *N. ceranae* in colony losses is difficult to define exactly, although it is the second most prevalent biological agent related to the decrease in worker bees. The differentiation and etiologic diagnosis of nosemosis is simple, but it requires qualified staff. It requires two steps: first, confirm the presence of spores by microscopy; second, use molecular methods to confirm the species. However, by the time the beekeeper detects visible symptoms, the colony is practically dead. Type C nosemosis, provoked by *N. ceranae* infection, has a complex pathogenesis, and it alters the physiology and behavior of individual bees and the whole hive. Both the immunosuppressive effect that these microsporidia produce, and the disturbance of the hive organization contribute to weakening it. The consequences are severe in several ways from an ecological, agronomic, and economic point of view. However, the data of mortality caused by *N. ceranae* infection are not clear, and it is necessary to know the real longevity data in infected colonies and understand and act on all causes involved. In the treatment of nosemosis, fumagillin has been widely used, although with negative effects on the metabolism of bees, so research on alternative treatments has become urgent. The utility of products such as nutraceuticals, plant extracts, probiotics, and veterinary drugs (Api-Bioxal^®^ and ApiHerb^®^) has been proven, with different results. For example, one nutraceutical, naringenin, seems to reduce the number of spores by 64%, whereas extracts of *Olea europaea* and *Laurus nobilis* inhibited the development of microsporidia. In this sense, combining different alternative treatments could be a good way to diminish bee losses due to type C nosemosis. However, other actions, such as improving the management of colonies and diminishing the use of insecticides, are necessary to increase bee populations worldwide. Finally, a “one-health” approach seems necessary to reverse the decrease of bee populations around the world.

## Figures and Tables

**Table 1 vetsci-09-00130-t001:** Efficacy of alternative treatments against *N. ceranae*.

Substance	Name	Efficacy	Reference
Plant extracts	*Laurus nobilis* (bay laurel)	Inhibition of *N. ceranae* development	[129]
	*Olea europaea* (olive)	Inhibition of *Nosema* spp. development in larvae and adult bees	[130]
	Oregano oil	40% reduction of *N. ceranae* spores	[111]
	Thymol	41% reduction of *N. ceranae* spores	[111]
Nutraceuticals	Sulforaphane	64% reduction of *N. ceranae* spores	[111]
	Naringenin	49% reduction of *N. ceranae* spores	[111]
	Carvacrol	57% reduction of *N. ceranae* spores	[111]
	Chitosan	>60% reduction of *N. ceranae* spores	[122]
	Peptidoglycan	>60% reduction of *N. ceranae* spores	[122]
Probiotics	*Bifidobacterium*	90% reduction of *N. ceranae* load and 47.7% reduction of infected bees	[112]
	*Lactobacillus* spp.	90% reduction of *N. ceranae* load and 47.7% reduction of infected bees	[112]
	*Parasaccharibacter apium*	56.8% reduction of *N. ceranae* spores	[114]
	Pentadecapeptide BPC 157	68% reduction of *N. ceranae* spores	[110]
Other compounds	Propolis	72% reduction of *N. ceranae* load in infected bees	[120]
Veterinary drugs	Api-Bioxal^®^	50% reduction of infected bees	[124]
	ApiHerb^®^	50% reduction of infected bees	[124]

## Data Availability

Not applicable.

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
