# Peer review of "The Role of Nosema ceranae (Microsporidia: Nosematidae) in Honey Bee Colony Losses and Current Insights on Treatment"

_vetsci, 2022, doi:10.3390/vetsci9030130_

Round 1

Reviewer 1 Report

The manuscript has summarized a good deal of references in this topic. However, some questions and mistakes need to be clarified or revised:

 L10:Abstract,” Honeybee populations have declined considerably in the last years”, where is the data shows the decline population in last year?

Table1: What does “Mixosporidia” mean? Is it misspelling of “Microsporidia”?

L37-39: In table 1, authors include all the pathogens without separating those high prevalent and rarely happened, it is misleading.   

L47:The sentence”The most prevalent biological agent related to the decrease of worker bees’ “is grammatically wrong. There should have a noun after the single quotation mark .

L49:The sentence”...mainly the species Nosema apis, N. ceranae and N. bombi, being the first two responsible for nosemosis in Apis spp.” Bumble bee belongs to Bumbus no Apis. so N. bombi is no a disease of Apis.

L119:”Major Royal Jelly Proteins”,There is not necessary in capital.

L154: “apidaecine, avaecine, hymenoptaecine and defensine” are wrongly spelled: No “e” in the end, and “avaecine” should be “abaecin”.

L231: “They also detected normal levels of trehalose in hypopharyngeal glands of infected bees.” ---The trehalose is the main carbohydrate in the hemolymph, and hypopharyngeal gland is the source of major royal jelly protein. I was confused by the “trehalose in hypopharyngeal gland”. After looking into the reference [120], its abstract clearly said “propolis treated bees had significantly higher haemolymph trehalose levels and hypopharyngeal gland protein content similar to levels of uninfected bees

”. I rather believe that the authors are careless on this sentence than lacking of common knowledge about honey bee physiology.

L248 : Table 2, in the second column,does “propolis” belong to “probiotics”? and, what belongs to “protein ” ?

L343: reference 22, “Apis Mellifera” -”M” should be in lower case. However, the similar mistakes are all over the references, like “Nosema Ceranae and Nosema Apis” ,Bombus Atratus and Bombus Bellicosus”(L448). and “Nosema Spp.” in L465.

Author Response

See document.

Reviewer 2 Report

I attached a file with my comments, which, if included, will make the manuscript better readable and understandable. Extensive English editing is required. I suggest consulting a native English speaker.

Author Response

see document.

Reviewer 3 Report

This review is very good and comprehensive.  It will be valuable to many others in the field.  Only a few minor issues:

line 87 should read "...cannot produce enough eggs and ..."

line 298 refers to "Rustic and environmentally friendly bees ..."  What does this mean?  Do the authors refer to feral colonies that are surviving without beekeeping assistacne?  Please clarify.

Author Response

Dear reviewer,

Following your recommendation, the sentence "the queen cannot produce enough pupae" has been changed to "the queen cannot produce enough eggs".

In the line 298, to avoid confusion, we have removed "rustic and environmentally friendly bees have better survival".

Round 2

Reviewer 2 Report

The Authors made all the changes I suggested and now the manuscript lokks ready to be published in my opinion.

Author Response

Many thanks for all.